# Estimated true prevalence and associated risk factors of bovine paratuberculosis antibodies in dairy herds in Bangladesh

Adel Md. Kibria[1‡], Md. Nazmul Islam[1‡], Md. Shaffiul Alam[1‡],
Bishwo Jyoti Adhikari[2], Shanta Islam[2], Abdullah Abu Rafeh[2],
R. S. Mahmud Hasan[1], Md. Siddiqur Rahman[1], A. K. M. Anisur Rahman[1*]

1 Epidemiology and Preventive Medicine Laboratory, Department of Medicine, Bangladesh Agricultural University, Mymensingh, Bangladesh, 2 Faculty of Veterinary Science, Bangladesh Agricultural University, Mymensingh, Bangladesh

‡ These authors contributed equally to this work and share first authorship.
* arahman_med@bau.edu.bd

## Abstract

Bovine paratuberculosis (PTB), caused by *Mycobacterium avium* subsp. *paratuberculosis* (MAP), is a chronic infection that undermines cattle productivity and farm profitability. In Bangladesh, no previous studies have estimated PTB prevalence or identified associated risk factors. This cross-sectional study (January 2023 to December 2024) was conducted across 14 districts and 138 dairy farms to estimate true MAP antibody prevalence and identify factors associated antibody detection. Bulk milk samples from all herds (n = 138) and 880 individual milk samples from ELISA-positive herds were tested using a commercial indirect ELISA. True herd- and cow-level antibody prevalence was estimated using Bayesian latent class models. Risk factors were identified using multivariable logistic regression (herd-level) and mixed-effects logistic regression (cow-level). The apparent antibody prevalence was 7.2% at the herd-level and 12.2% at the cow-level, with herd-level prevalence ranging from 0.9% to 37%. After adjusting for test sensitivity and specificity, true antibody prevalence estimates decreased to 4.9% (herd level) and 9.4% (cow level). Herds with more than two retained placenta cases had significantly higher odds of MAP antibody detection (odds ratio [OR] = 9.42) than herds with two or fewer such events. At the cow level, older age (>7.9 years) was associated with higher odds of antibody positivity compared with cows ≤4 years. Cows with anestrus (OR = 3.89) or repeat breeding (OR = 2.63) had significantly higher odds of MAP antibody detection compared with cows without these disorders. Milk antibody levels showed a negative association with milk yield, with each unit increase in the sample-to-positive ratio corresponding to a 0.018 L/day reduction. These findings indicate that apparent antibody prevalence substantially overestimates PTB exposure in the study areas. The associations with retained placenta, anestrus, and repeat breeding suggest potential

**Data availability statement:** All relevant data are within the paper and its Supporting Information files.

**Funding:** This research was funded by Livestock and Dairy Development Project, grant number 2022/7/LDDP. The funders had no role in study design, data collection and analysis, decision to publish, or preparation of the manuscript.

**Competing interests:** The authors have declared that no competing interests exist.

reproductive impacts that warrant further investigation. Given the purposive sampling strategy and reliance on milk ELISA, the results should be interpreted within the context of the study population. Enhancing diagnostic capacity and integrating PTB considerations into reproductive herd management may help reduce productivity losses, while larger, probability-based studies are needed to assess the national burden.

## Introduction

Bovine paratuberculosis (Johne's disease, PTB) is a chronic, incurable, and slowly progressive enteric infection of ruminants caused by *Mycobacterium avium* subspecies *paratuberculosis* (MAP). This pathogen is a slow-growing, acid-fast, intracellular bacterium that primarily targets the ileum and associated lymphatic tissues, resulting in granulomatous enteritis and chronic wasting in infected animals [1,2]. Infected animals often remain subclinical for years, intermittently shedding MAP in feces, milk and colostrum [3,4], which facilitates herd-level transmission by the fecal–oral route (e.g., calves ingesting contaminated colostrum or feed). Clinically, PTB causes major production losses: affected cattle exhibit reduced milk yield, poor fertility and stunted growth, making PTB one of the most costly infectious diseases of dairy cattle [2,4]. In addition to animal health impacts, the disease imposes considerable economic costs through premature culling, reduced productivity, and increased veterinary expenses, with global losses estimated in the hundreds of millions of dollars annually [5,6]. MAP is also considered a potential zoonotic threat: viable bacilli have been detected in raw and pasteurized milk and in intestinal biopsies of Crohn's disease patients, raising concerns about its possible role in human inflammatory bowel disease [7–9].

Paratuberculosis has a truly global distribution. First described in Europe in the late 19th century, the disease has now been reported on every inhabited continent [2,9]. A review of 48 countries found that paratuberculosis is widespread, with more than 20% of livestock herds testing positive in about half of them [2]. Reported animal-level prevalence varies widely, from low single digits to 30–40% in endemic regions [2,10]. However, subclinical infection often goes undetected: clinical signs are nonspecific, and available diagnostic tests have limited sensitivity in early stage of the infection, meaning the true burden is likely underestimated [2,4]. Formal control programs exist in only a few—mostly developed—countries, while 76% of countries without MAP control programs are in Asia, Africa, or South America [2]. This lack of surveillance leaves the paratuberculosis status and prevalence of many countries uncertain, particularly in Asia.

Within South Asia, several studies confirm that MAP infection is already established in cattle and buffalo populations. For example, a survey of dairy farms in Nepal detected MAP DNA in 16.6% of cattle using fecal PCR (polymerase chain reaction) [11]. In Pakistan, serological screening of public dairy farms using the tuberculin skin test, ELISA, and PCR identified *Mycobacterium avium* subsp. *paratuberculosis* (MAP) in 3.8% of individual cattle and buffaloes [12]. In India, MAP exposure is considerable, with animal-level ELISA seroprevalence commonly reported in the 20–40% range. A study in North India recorded an overall seropositivity of about 29% (28.6%

in buffalo, 29.8% in cattle), with state-level variation ranging from 31.9% in Uttar Pradesh to 23.3% in Punjab [3,13]. Similarly, in eastern India, a 2017 study in West Bengal reported 19.9% herd-level MAP positivity in organized farms using a combined ELISA and delayed-type hypersensitivity test [14]. Collectively, these studies indicate that MAP is widely present in South Asian ruminants, though prevalence varies by geography, diagnostic method, and management practices.

In contrast, in Bangladesh, to the best of our knowledge, there is no published report on the antibody prevalence or risk factors of bovine paratuberculosis, highlighting a critical knowledge gap. The country maintains approximately 24.4 million cattle and 1.5 million buffalo [15], most raised in smallholder and peri-urban systems characterized by low biosecurity. Given the high density of ruminants and their central role in rural livelihoods and national food security, undetected PTB could have serious health and economic consequences. Global experts have already emphasized that many countries—particularly in Asia—remain "unknown" for MAP prevalence due to insufficient surveillance [2].

To address this gap, we conducted a cross-sectional study of dairy herds across multiple regions of Bangladesh. Our objectives were to estimate the herd- and cow-level true antibody prevalence of MAP in cattle and to identify associated risk factors at both levels.

## Materials & methods

### Ethics statement

The study was conducted in accordance with the Declaration of Helsinki. The animal study protocol was approved by the Animal Welfare and Experimentation Ethical Committee (AWEEC) of Bangladesh Agricultural University (AWEEC/BAU/2022/07).

### Study design, study and target population

A cross-sectional study was conducted from January 2023 to December 2024 across 14 districts of Bangladesh (Fig 1). The target population comprised dairy cattle in Bangladesh, with the study population specifically drawn from dairy-dense regions. Although a probabilistic national sampling framework was not feasible due to the absence of a comprehensive herd registry, deliberate efforts were made to ensure broad geographic representation. Given the higher likelihood of infectious diseases in larger herds, the study prioritised medium-to-large herds; however, as this is the first milk-based investigation of its kind in Bangladesh, herds with fewer than 10 lactating cows were also included to reflect typical herd-size variation. The 14 study districts were purposively selected based on predefined epidemiological and operational criteria. In the absence of a national dairy herd sampling frame in Bangladesh, districts were chosen from established dairy-dense regions to ensure coverage of areas with the highest cattle populations and potential for disease transmission. Selection was guided by (i) documented dairy production intensity, (ii) the presence of medium- to large-scale herds, and (iii) accessibility for timely field operations. Within each district, herds were identified using records from the Department of Livestock Services (DLS) and through consultation with local veterinary authorities and field staff. Milk sampling was conducted by veterinary professionals and trained personnel to ensure consistent procedures and high data quality.

### Sampling size calculation and sampling protocol

The sample size of the study was calculated using the equation (1);

$$n = \frac{Z^2 P(1-P)}{d^2}$$

(1)

In this equation, Z is the Z-score for a 95% confidence level, which is 1.96. P represents the expected prevalence, set at 50.0% or 0.50, and d is the precision, determined to be 3.5% or 0.035. Based on these assumptions, the calculated sample size was 784. However, a total of 880 samples were ultimately collected to increase the statistical power of the

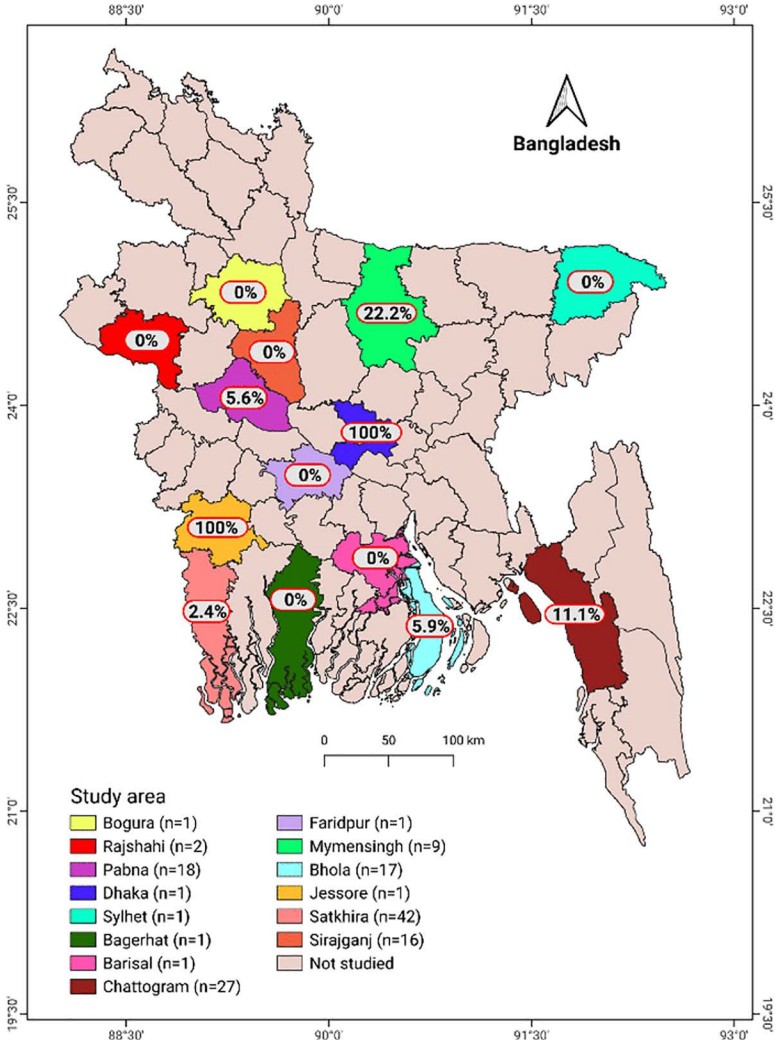

**Fig 1. Map of Bangladesh showing the 14 study districts and herd-level antibody prevalence of bovine paratuberculosis.** The value n represents the number of herds sampled in each district, rather than the number of individual animals. In the districts where MAP antibody was detected, prevalence varied widely, with herd-level seropositivity ranging from 2.4% to 100%. The map was generated using QGIS (version 3.36.1) with district boundary shapefiles obtained from the GADM database [26].

study and enhance the precision of estimates, particularly for subgroup analyses. For individual-level sampling, bulk milk samples were first collected from 138 purposively selected farms and screened using a commercial indirect ELISA. Herds were classified as positive when the bulk milk S/P ratio exceeded the manufacturer's recommended cutoff (ID Screen® Paratuberculosis Indirect, Innovative Diagnostics, Grabels, France). From the ten herds that tested ELISA-positive, a sampling frame was constructed by listing all lactating cows. A total of 880 cows were then randomly selected from this frame using the 'RANDBETWEEN' function in Microsoft Excel to ensure unbiased selection. In districts where more than one positive herd was eligible, inclusion was further guided by herd owner willingness and logistical feasibility.

## Collection of milk samples

Approximately 40 mL of bulk milk was collected from each herd. In addition, 40 mL of fresh milk was aseptically collected from each cow into sterile Falcon tubes. All tubes were clearly labeled with the corresponding animal and herd

identification number. Samples were immediately frozen, transported to the Department of Medicine, Bangladesh Agricultural University in an icebox, and stored at –20 °C until further analysis.

### Herd and individual-level data collection

Data on potential risk factors were collected using pretested questionnaires at both herd and individual levels. Herd-level variables included herd size, numbers of lactating cows, heifers, and calves, annual abortions, retained fetal membranes, repeat breeding cases, daily milk yield, and semen source. As all participating herds practiced zero-grazing and were managed exclusively indoors, inter-herd contact was not applicable, and reliable records on animal purchase or trading were unavailable. Therefore, these variables were not included in the study. Individual-level variables included age, breed, body condition, pregnancy status, history of abortion, retained fetal membranes, reproductive disorders, number of calves, lactation stage, daily milk yield, and lifetime mastitis history.

### Processing milk samples

Milk samples were thawed at room temperature until cream separated from lactoserum. The lactoserum was transferred into 1.5 mL Eppendorf tubes and loaded into a 96-well plate, with wells A1 and B1 containing negative controls and C1 and D1 containing positive controls. The remaining wells were filled with lactoserum for subsequent transfer to the ELISA plate using a 12-channel micropipette.

### Indirect enzyme linked immunosorbent assay (iELISA)

Serum and milk samples were tested for antibodies against *Mycobacterium avium* subsp. *paratuberculosis* using the ID Screen® Paratuberculosis Indirect ELISA kit (PARAS-10P, Innovative Diagnostics, Grabels, France) following the manufacturer's protocol. Reagents stored at 2–8 °C were equilibrated to room temperature (21 °C ± 5 °C) and homogenized by vortexing. Negative (10 µL) and positive (10 µL) controls were diluted with 110 µL of Dilution Buffer 6 in wells A1–B1 and C1–D1, respectively. Remaining wells received 80 µL of sample and 80 µL of Dilution Buffer 6. Plates were covered and incubated at 21 °C ± 5 °C for 5–45 min. Subsequently, 100 µL of prepared controls and samples were transferred to coated ELISA microplates and incubated for 45 ± 4 min at 21 °C ± 5 °C. Wells were washed three times with ≥300 µL wash solution without drying between washes. Conjugate was prepared at a 1:10 dilution in Dilution Buffer 3; 100 µL was added per well, followed by incubation for 30 ± 3 min at 21 °C ± 5 °C and three further washes. Substrate solution (100 µL) was added and incubated for 15 ± 2 min in the dark at 21 °C ± 5 °C. The reaction was stopped with 100 µL Stop Solution, and optical density was read at 450 nm. Wash solution was prepared by diluting Wash Concentrate (1X) in distilled water (1:20) at room temperature.

### Test validation

The test was considered valid if the mean optical density of the positive control ($OD_{PC}$) exceeded 0.350 and the ratio of ($OD_{NC}$) to the mean negative control optical density ($OD_{NC}$) exceeded 3. The sample-to-positive ratio (S/P%) was calculated as:

$$S/P\% = \frac{OD_{sample} - OD_{NC}}{OD_{PC} - OD_{NC}} \times 100$$

Samples were considered positive if S/P% > 15% for bulk milk and > 30% for individual samples.

### Data analysis

**Descriptive statistics.** Numeric data, including OD values, SP%, age, body weight, and parity, were summarized using the `summary` function in R version 4.5.1 [16]. The prevalence of paratuberculosis and its distribution across

variable categories were calculated using the `tabpct` function from the `epiDisplay` package [17]. Data visualizations were generated using the `ggplot` function from the `ggplot2` package [18]. MAP antibody prevalence with 95% confidence intervals was calculated using the `prop.test`.

### True antibody prevalence estimation

The true MAP antibody prevalence at the herd and cow levels was estimated using a Bayesian model, as described earlier [19]. True herd-level antibody prevalence was estimated using the iELISA's reported sensitivity (63%, 95% CI: 41–81%) and specificity (92%, 95% CI: 86–96%) [20]. Cow-level prevalence estimates used the commercial ELISA's diagnostic accuracy (Se: 85.7%, 95% CI: 76.4–95.1%; Sp: 95.1%, 95% CI: 92.6–97.7%) [21]. Sensitivity and specificity priors were specified as Beta distributions derived from published means and confidence intervals using the beta.select() function from the 'LearnBayes' package [22]. The model was implemented in the 'R2jags' package in R version 4.5.1, with three parallel Markov Chain Monte Carlo (MCMC) chains. After a burn-in of 50,000 iterations, estimates were obtained from an additional 500,000 iterations with a thinning interval of 10. Convergence was evaluated using trace plots and Gelman–Rubin diagnostics [23]. The R code used to estimate true prevalence at both herd and cow levels is provided in the supplementary file 5.

### Identification of risk factors

**Herd-level risk factors.** A herd was considered MAP-positive if its bulk milk tested positive for MAP-specific antibodies using the commercial ELISA assay. Continuous herd-level variables were categorized based on quartiles or median values to facilitate analysis. Initially, univariable logistic regression analyses were conducted with herd-level MAP seropositivity as the outcome and each potential risk factor as a predictor. Variables showing a p-value ≤0.20 were included in the multivariable model. Multicollinearity among the selected variables was evaluated using the 'vif()' function in R (version 4.5.1), with a generalized variance inflation factor (GVIF) >2 considered indicative of multicollinearity [24]. The final multivariable model was built using a stepwise logistic regression approach, and confounding and interaction terms were assessed using standard methods [25].

**Cow-level risk factors.** Individual cows were classified as MAP-positive if their milk samples tested positive by ELISA. Continuous cow-level variables, including age, lactation stage, parity, and milk yield, were converted into categorical variables based primarily on quartiles and occasionally medians. This approach avoids arbitrary cut-offs and reflects the distribution within our study population, ensuring balanced groupings for interpretability. To assess individual-level risk factors, univariable mixed-effects logistic regression was performed for each predictor, with cow-level MAP seropositivity as the outcome and farm included as a random intercept to account for clustering within herds. Predictors with p ≤ 0.20 were selected for inclusion in the multivariable mixed-effects model. Multicollinearity among variables was assessed using the same method as for herd-level factors [24]. The final model was constructed using stepwise forward multivariable mixed-effects logistic regression, with confounding and interaction checked following established procedures [24]. The datasets used to identify herd- and cow-level risk factors are provided in Supplementary files 1 and 2, respectively. The R code used to identify risk factors for MAP antibody detection at both herd and cow levels were provided as supplementary files 3 and 4.

## Results

### Descriptive statistics

**Herd.** The study was carried out on 138 dairy herds throughout 14 districts in Bangladesh (Fig 1). The herd size ranged from 4 to 2850 animals, with a mean of 119. The mean number of lactating cows in these herds ranged from 2 to 1300, with a mean of 37. With a range of 0–32, the mean number of abortions per herd was 2. Retained fetal membrane

cases ranged from 0 to 70, with a mean of 4. For each herd, there was a mean of 2 anestrus cases (range: 0–37) and 10 repeat breeding cases (range: 0–600). The mean yield of milk per herd was 216.12 kg per day, with a range of 8–6400 kg.

**Cow.** In total, 880 milk samples were collected from MAP antibody-positive herds located in 4 of the 14 surveyed districts. The cows had a mean age of 5.69 years, ranging from 1.4 to 15.7 years. On average, each cow had 3 calves (range: 1–10). The mean lactation period was 545 months (ranging between 1 and 11 months), while the mean body weight was 341.7 kg (range: 130–800 kg). The cows produced a mean daily milk yield of 10.14 kg, with values ranging from 1 to 30 kg.

### Herd and cow-level apparent MAP antibody prevalence

The overall apparent herd-level MAP antibody prevalence was 7.2% (95% CI: 3.7–13.3%). Among cows sampled from seropositive herds, the apparent antibody prevalence was 12.2% (95% CI: 10.0–15.0%). Prevalence varied markedly across herds, ranging from 0.9% to 37% (Fig 2).

### True MAP antibody prevalence

The estimated herd-level true MAP antibody prevalence was 4.9% (95% credible interval (CrI): 0.2%–15%) (Fig 3), compared with an apparent prevalence of 7.2%. At the cow level, the estimated cow-level true antibody prevalence of MAP was 9.4% (95% CrI: 5.2%–13.4%) (Fig 4).

### Association between SP and milk yield

A significant negative association was observed between SP% and milk yield (Fig 5). Each one-unit increase in SP was associated with a 0.018 L/day decrease in yield (β = −0.018, SE = 0.004, t = −4.49, p < 0.001). The model intercept was 9.87 L/day. Residuals ranged from −8.86 to 20.54 L/day with a median near zero, and the residual standard error was 5.06 L/day. The model explained little variation in yield ($R^2$ = 0.016; adjusted $R^2$ = 0.015).

### Risk factor

**Herd.** In the univariable analysis, herd size, number of calves, abortion history, retained placenta, average milk yield, and repeat breeding were associated with MAP antibody prevalence at $p \leq 0.20$ and were therefore considered

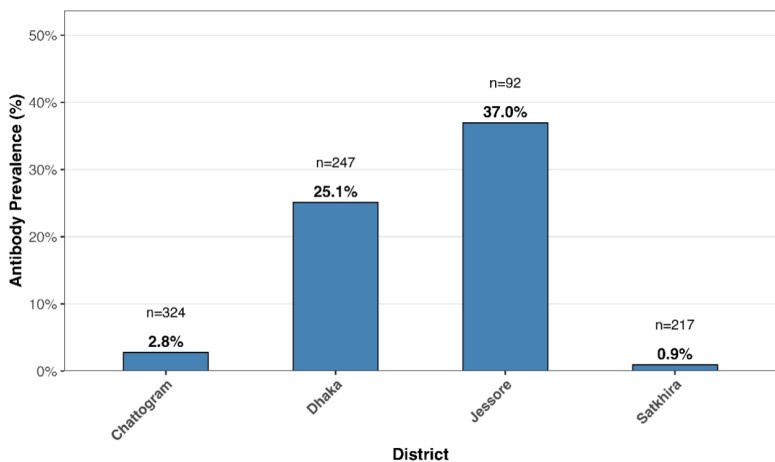

**Fig 2. Apparent antibody prevalence of paratuberculosis in individual cows within ELISA-positive herds.** For confidentiality, each district represents one herd, and herd names are not shown; district labels are used as proxies for individual herds.

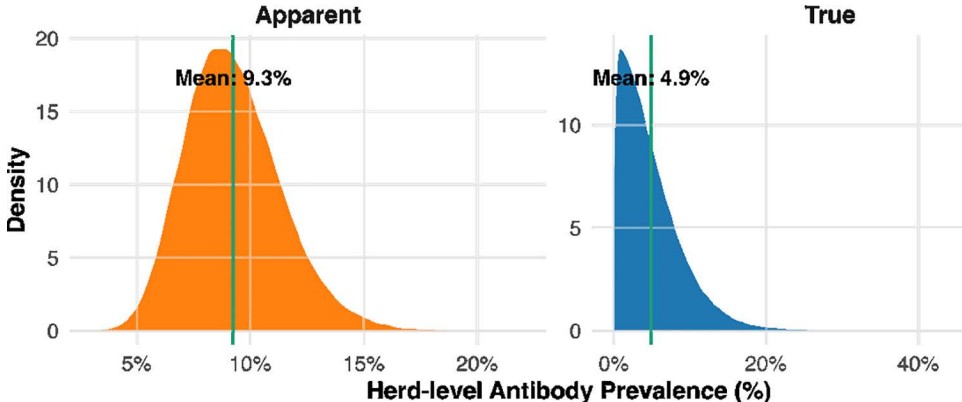

**Fig 3. Posterior distributions of apparent and true herd-level antibody prevalence of MAP in dairy herds of Bangladesh.** The vertical green line indicates the mean prevalence for each distribution.

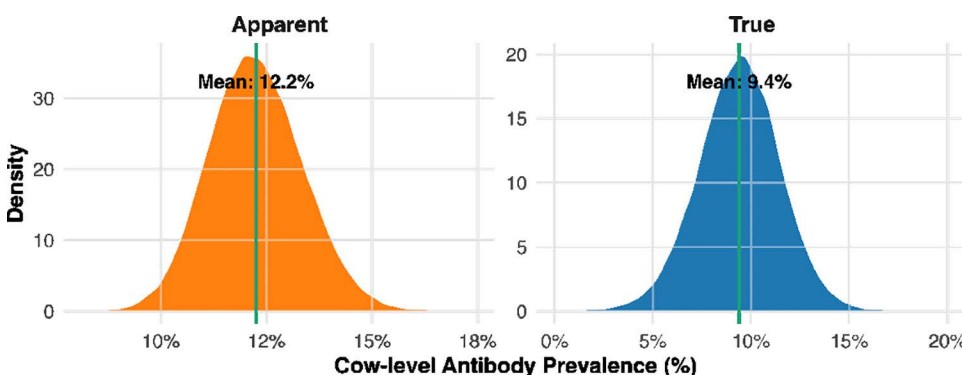

**Fig 4. Posterior distributions of apparent and true cow-level antibody prevalence of MAP in dairy cows from positive herds in Bangladesh.** The vertical green line indicates the mean prevalence for each distribution.

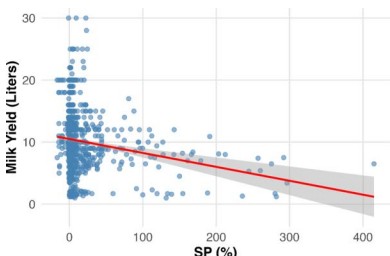

**Fig 5. Relationship between sample to positive (SP%) and milk yield in dairy cows.** The scatterplot illustrates the negative association between SP and milk yield, with the regression line showing the fitted linear trend.

for inclusion in the multivariable logistic regression model (Table 1). No multicollinearity was detected among these variables. In the multivariable model, retention of placenta remained significantly associated with MAP antibody positivity. Herds with more than two retained placenta events had significantly higher odds of MAP antibody positivity (OR = 9.42, 95% CI: 1.12–79.13, p = 0.04) compared with those with two or fewer. In the crude model, the odds ratio

**Table 1. Univariable associations between herd-level *Mycobacterium avium* subsp. *paratuberculosis* antibody detection and explanatory variables.**

| Variables | Category | Tested | Positive (%) | OR (95% CI) | P-value (AIC) |
|---|---|---|---|---|---|
| **Herd size** | | | | | 0.08 (73.11) |
| | ≤ 13 | 37 | 1 (2.7) | Reference | |
| | > 13 – 24 | 32 | 1 (3.1) | 1.16 (0.07 – 19.35) | |
| | > 24 – 70 | 35 | 2 (5.7) | 2.18 (0.19 – 25.19) | |
| | > 70 | 34 | 6 (17.6) | 7.71 (0.88 – 67.82) | |
| **Lactating cow number** | | | | | 0.06 (71.23) |
| | ≤ 12 | 71 | 2 (2.8) | Reference | |
| | > 12 | 67 | 8 (11.9) | 4.68 (0.96 – 22.89) | |
| **Number of calves** | | | | | 0.07 (72.49) |
| | ≤ 4 | 79 | 3 (3.8) | Reference | |
| | > 4 | 59 | 7 (11.9) | 3.41 (0.84 – 13.8) | |
| **Abortion case** | | | | | 0.02 (70.19) |
| | <=2 | 79 | 4 (3.9) | Reference | |
| | > 2 | 59 | 6 (16.7) | 4.9 (1.3 – 18.52) | |
| **Milk yield (Kg)** | | | | | 0.12 (73.14) |
| | ≤ 110 | 75 | 3 (4) | Reference | |
| | > 110 | 63 | 7 (11.1) | 3 (0.74 – 12.13) | |
| **Retention of Placenta** | | | | | 0.01 (65.12) |
| | ≤ 2 | 79 | 1 (1.3) | Reference | |
| | > 2 | 59 | 9 (15.3) | 14.04 (1.73 – 114.23) | |
| **Repeat Breeding case** | | | | | 0.03 (67.89) |
| | ≤ 3 | 69 | 2 (2.9) | Reference | |
| | > 3 | 69 | 8 (11.6) | 10.2 (1.26 – 82.88) | |
| **Semen source** | | | | | 0.44 |
| | Source 1 | 8 | 2 (25) | 7.33 (0.56 – 95.28) | |
| | Source 2 | 4 | 1 (25) | 7.33 (0.36 – 150.68) | |
| | Source 3 | 23 | 1 (4.3) | Reference | |
| | Source 4 | 28 | 0 (0) | Not analyzed | |
| | Source 5 | 47 | 4 (8.5) | 2.05 (0.22 – 19.43) | |
| | Source 6 | 1 | 1 (100) | Not analyzed | |
| | Source 7 | 5 | 1 (20) | 5.5 (0.28 – 107.13) | |
| | Source 8 | 22 | 0 (0) | Not analyzed | |
| **Overall** | | 138 | 10 (7.2; 95% CI: 3.7–13.3) | | |

Semen source coded to maintain confidentiality of AI providers; OD: Odds ratio; CI: Confidence Interval.

was 14.04 (95% CI: 1.73–114.23), and adjustment for repeat breeding reduced this estimate by about 33%. Although repeat breeding (> 3 cases) was not statistically significant (OR = 6.08, 95% CI: 0.72–51.62, p = 0.09), its inclusion demonstrated a confounding effect, highlighting its influence on the relationship between retained placenta and MAP antibody positivity (Table 2).

**Cow.** In the univariable screening, age, reproductive disorders, and pregnancy were associated with MAP antibody prevalence at $p \leq 0.20$ (Table 3). No multicollinearity was detected among explanatory variables. In the final mixed-effects multivariable logistic regression, no two-variable model was statistically significant. Among the univariable models, reproductive disorders provided the best fit based on *p*-value and AIC. Cows with anestrus had 4.18 times higher odds of

**Table 2. Herd-level factors associated with *Mycobacterium avium* subsp. *paratuberculosis* antibody detection in the final multivariable model.**

| Variables | Category | Estimate (SE) | Odds ratio (95% Confidence Interval) | P-value |
|---|---|---|---|---|
| **Retention of Placenta** | | | | |
| | ≤ 2 | – | Reference | |
| | > 2 | 2.24 (1.09) | 9.42 (1.12, 79.13) | 0.04 |
| **Repeat Breeding case (confounder)** | | | | |
| | ≤ 3 | – | Reference | |
| | > 3 | 1.81 (1.09) | 6.08 (0.72, 51.62) | 0.09 |

AIC for two variable model: 63.15.

MAP antibody positivity (95% CI: 2.01–8.71; $p<0.001$), and cows with repeat breeding had 2.92 times higher odds (95% CI: 1.68–5.09; $p<0.001$) compared to cows without reproductive disorders (Table 3).

Age category >7.9 years was associated with higher odds of testing positive compared with cows ≤4 years (OR = 2.34, 95% CI: 1.12–4.93, p = 0.02). Among reproductive disorders, cows with anestrus (OR = 3.89, 95% CI: 1.84–8.23, p < 0.001) or repeat breeding (OR = 2.63, 95% CI: 1.49–4.66, p < 0.001) had significantly higher odds of MAP antibody detection compared with cows without these disorders (Table 4).

## Discussion

This study provides the first comprehensive estimate of bovine MAP antibody prevalence and associated risk factors in Bangladesh. Our results reveal a substantial gap between apparent and true antibody prevalence, with true herd- and cow-level prevalence considerably lower than apparent estimates after adjusting for diagnostic test performance. These findings underscore the necessity of accounting for test characteristics when estimating disease burden, to ensure accurate and effective control strategies.

Using the iELISA's reported sensitivity (63%) and specificity (92%), we estimated the herd-level true antibody prevalence of MAP at 4.9%, consistent with reports from other low-endemicity regions. For example, Swiss dairy herds showed a true seroprevalence of 3.6% [27] while Canadian cow–calf herds reported 3% in eastern and 1% in western regions [28]. In our study, the true prevalence was lower than the apparent prevalence of 7.2%, illustrating the "inversion phenomenon" characteristic of low-prevalence settings. When true infection levels are low and test specificity is imperfect, false positives can outnumber true positives, causing adjusted prevalence to fall below observed proportions. Under the Rogan–Gladen correction, if apparent prevalence is lower than the probability of a false positive (1 − specificity), the resulting estimate may even approach zero or become negative [29]. This counterintuitive outcome highlights the importance of adjusting for test performance: apparent prevalence alone can overestimate infection burden, whereas adjusted estimates more accurately reflect MAP status in herds.

Consistent with herd-level findings, the cow-level true MAP antibody prevalence was 9.4%, lower than the apparent prevalence of 12.2%. Adjustment incorporated the diagnostic accuracy of the commercial ELISA (Se: 85.7%; Sp: 95.1%), demonstrating how imperfect specificity can produce false positives in low-prevalence populations. Our cow-level estimates align with other low- to moderate-endemicity regions. For instance, Northern Ireland reported a median animal-level true prevalence of 9.5% in dairy herds [30]. In China, MAP prevalence varied geographically, with southern regions showing only 2% infection and eastern regions reaching 12% [31].

In contrast, studies from the Indian subcontinent suggest similar or higher apparent prevalence, although true prevalence was rarely estimated. In Pakistan, MAP was detected in 3.8% of individual cattle and buffaloes [12]. n India, ELISA-based seroprevalence commonly ranged from 20–40%, with one North Indian study reporting 29% overall (28.6% in buffalo, 29.8% in cattle) and state-level variation from 23.3% to 31.9% [3,13]. These findings emphasize that apparent prevalence should be interpreted cautiously, as it may overestimate true infection burden.

**Table 3. Results of the univariable analysis between cow-level *Mycobacterium avium* subsp. *paratuberculosis* antibody detection and explanatory variables.**

| Variables | Category | Tested | Positive (%) | OR (95% CI) | *P*-value (AIC) |
|---|---|---|---|---|---|
| **Age (years)** | | | | | 0.02 (521.9) |
| | Up to 4 | 267 | 13 (4.9) | Reference | |
| | 4 to 5.3 | 198 | 29 (14.6)) | 2.03 (0.98 −4.21) | |
| | 5.3 to 7.9 | 213 | 28 (13.1) | 1.97 (0.95- 4.06) | |
| | > 7.9 | 202 | 37 (18.3) | 3.19 (1.56- 6.53) | |
| **Breed** | | | | | 0.21 |
| | Indigenous | 75 | 13 (17.3) | 1.55(0.76- 3.14) | |
| | Cross | 805 | 94 (11.7) | Reference | |
| **Physical condition** | | | | | 0.21 |
| | Normal | 706 | 77 (10.9) | Reference | |
| | Thin | 174 | 30 (17.2) | 1.18 (0.64 −2.16) | |
| **Pregnancy** | | | | | 0.19 (527.2) |
| | No | 554 | 60 (10.8) | Reference | |
| | Yes | 326 | 47 (14.4) | 1.56 (0.79-3.06) | |
| **Reproductive disorders** | | | | | <0.001 (507.3) |
| | Anestrus | 179 | 25 (14) | 4.18 (2.01- 8.71) | |
| | No | 447 | 23 (5.1) | Reference | |
| | Repeat breeding | 254 | 59 (23.2) | 2.92 (1.67-5.09) | |
| **Abortion** | | | | | 0.42 |
| | No | 849 | 106 (12.5) | 2.32 (0.29- 18.22) | |
| | Yes | 31 | 1 (3.2) | Reference | |
| **Retention of Placenta** | | | | | 0.24 |
| | No | 790 | 99 (12.5) | 1.65 (0.70−0 3.89) | |
| | Yes | 90 | 8 (8.9) | Reference | |
| **Number of Calves** | | | | | 0.22 |
| | ≤ 2 | 511 | 63 (12.3) | 0.75 (0.48- 1.187) | |
| | > 2 | 369 | 44 (11.9) | Reference | |
| **Lactation Stage (months)** | | | | | 0.23 |
| | Up to 3 | 295 | 17 (5.8) | Reference | |
| | 3–5 | 170 | 22 (12.9) | 0.96 (0.46-1.98) | |
| | 5–8 | 237 | 40(16.9) | 1.46 (0.75- 2.85) | |
| | > 8 | 178 | 28(15.7) | 0.82 (0.370- 1.80) | |
| **Milk Yield (Kg)** | | | | | 0.84 |
| | ≤ 9 | 481 | 78 (16.2) | 0.95 (0.55-1.64) | |
| | > 9 | 399 | 29 (7.3) | Reference | |
| **Mastitis** | | | | | 0.42 |
| | No | 746 | 101(13.5) | 1.46 (0.58-3.67) | |
| | Yes | 134 | 6 (4.5) | Reference | |
| **Overall prevalence** | | 880 | 107 (12.2) | | |

OD: Odds ratio; CI: Confidence Interval.

**Table 4. Cow-level factors associated with _Mycobacterium avium_ subsp. _paratuberculosis_ antibody detection in the final multivariable model.**

| Variables | Category | Estimate (SE) | OR (95% CI) | _P_-value |
|---|---|---|---|---|
| **Age (years)** | | | | |
| | Up to 4 | – | Reference | – |
| | 4 to 5.3 | 0.40 (0.38) | 1.49 (0.70–3.19) | 0.29 |
| | 5.3 to 7.9 | 0.31 (0.38) | 1.37 (0.65–2.91) | 0.41 |
| | > 7.9 | 0.85 (0.37) | 2.34 (1.12–4.93) | 0.02 |
| **Reproductive disorders** | | | | |
| | Anestrus | 1.35 (0.38) | 3.89 (1.84–8.23) | <0.001 |
| | No | – | Reference | – |
| | Repeat breeding | 0.96 (0.29) | 2.63 (1.49–4.66) | <0.001 |

OD: Odds ratio; CI: Confidence Interval.

We observed a modest but statistically significant negative association between ELISA sample-to-positive (S/P) ratios and daily milk yield, with each one-unit increase in S/P corresponding to a 0.018 L/day reduction. Although small relative to other production determinants such as parity, lactation stage, and nutrition, this trend is consistent with previous reports showing reductions of approximately 1,285 kg milk over five lactations in ELISA-positive cows [32] and declines in herds with >10% antibody prevalence, particularly among older cows, along with reduced milk components [33]. Other studies, however, reported no effect or even higher production in MAP-positive animals [34,35]. These inconsistencies likely reflect differences in diagnostic methods, positivity thresholds, herd management, and infection stage. Overall, our findings indicate an association between antibody levels and milk yield rather than a causal effect, highlighting broader uncertainty regarding MAP's impact on milk production.

Our finding that herds with more than two retained placenta events had significantly higher odds of MAP antibody positivity (OR = 9.42) points to a potentially underexplored association between MAP infection and impaired placental expulsion. Although studies directly linking MAP to retained placenta in cattle are limited, this association is biologically plausible, as MAP has been detected in reproductive tissues [36,37].

Older cattle in our study were more likely to test positive for MAP antibodies, consistent with findings from other countries, where seroprevalence generally increases with age, especially in animals older than three to four years [38–40]. Cows older than 7.9 years in our study had significantly higher odds of seropositivity compared with those ≤4 years. Although age thresholds vary across studies, these differences likely reflect herd structure, culling practices, infection pressure, and age-dependent ELISA sensitivity. Overall, our results reinforce the well-established trend of rising MAP antibody positivity with age, emphasizing the need to interpret age effects within herd-specific epidemiology and management contexts.

MAP antibody prevalence was also significantly associated with reproductive disorders. Anestrous cows had over four times the odds of testing positive, and repeat breeders nearly three times the odds, compared with reproductively healthy animals. These observations align with previous studies reporting longer calving intervals, delayed conception, and higher culling rates among MAP-positive cows [35,41]. While biologically plausible, these associations are not necessarily causal, reflecting the systemic effects of MAP on energy metabolism and immune function. Longitudinal studies are needed to clarify whether MAP infection directly impairs reproductive performance.

Several limitations should be acknowledged. First, the survey was not statistically balanced across Bangladesh, and herd sizes varied widely, which restricts the generalizability of the findings. Second, reliance solely on milk samples—an imperfect matrix for MAP detection—may have reduced diagnostic sensitivity. Although we adjusted prevalence estimates using the reported sensitivity and specificity of the ELISA, some uncertainty remains, as test performance may differ

under local field conditions, and imperfect specificity can disproportionately influence estimates in low-prevalence settings. The cross-sectional design also limits the ability to infer causality for associations with milk yield, age, and reproductive disorders. Finally, the extreme herd-level prevalence values observed in certain provinces are likely artefacts of sampling design rather than true geographic differences. These considerations are important for appropriate interpretation of the results.

## Conclusion

This study provides the first comprehensive assessment of bovine paratuberculosis antibody prevalence in Bangladesh. True herd- and cow-level MAP prevalence was lower than apparent prevalence after accounting for test performance, emphasizing the importance of diagnostic adjustment in low-prevalence settings. Older age and reproductive disorders were significantly associated with MAP seropositivity, while higher antibody levels showed a modest negative association with milk yield. These results contribute to understanding the epidemiology of MAP in Bangladesh and highlight the need for careful interpretation of apparent prevalence, as well as targeted disease control strategies that consider herd management and diagnostic limitations. Further longitudinal and larger-scale studies are needed to confirm these associations and inform effective interventions.

## Supporting information

**S1 File. Herd-level data to identify risk factors (Herd_data.csv).**
(CSV)

**S2 File. Cow-level data to identify risk factors (Cow_data.csv).**
(CSV)

**S3 File. R code for herd-level risk factor analysis (R_code_herd_level_risk_factors.txt).**
(TXT)

**S4 File. R code for cow-level risk factor analysis (R_code_cow_level_risk_factors.txt).**
(TXT)

**S5 File. R code to estimate herd and cow level true antibody prevalence (R_code_herd_cow_level_true_prevalence.txt).**
(TXT)

## Acknowledgments

The authors are grateful to the dairy farmers for participating in this study and for providing milk samples and data.

## Author contributions

**Conceptualization:** Md. Siddiqur Rahman, A. K. M. Anisur Rahman.

**Data curation:** Adel Md. Kibria, Md. Nazmul Islam, Shanta Islam, Abdullah Abu Rafeh, RS Mahmud Hasan.

**Formal analysis:** Md. Nazmul Islam, Bishwo Jyoti Adhikari, Shanta Islam.

**Funding acquisition:** Md. Siddiqur Rahman, A. K. M. Anisur Rahman.

**Investigation:** Adel Md. Kibria, Md. Shaffiul Alam, Bishwo Jyoti Adhikari.

**Methodology:** Adel Md. Kibria, Md. Nazmul Islam, Md. Shaffiul Alam, Bishwo Jyoti Adhikari, Shanta Islam.

**Project administration:** Md. Siddiqur Rahman, A. K. M. Anisur Rahman.

**Resources:** Md. Siddiqur Rahman.

**Software:** Md. Nazmul Islam, Bishwo Jyoti Adhikari, Shanta Islam, A. K. M. Anisur Rahman.

**Supervision:** A. K. M. Anisur Rahman.

**Validation:** Md. Shaffiul Alam, Bishwo Jyoti Adhikari, Abdullah Abu Rafeh, RS Mahmud Hasan.

**Visualization:** Md. Nazmul Islam, Shanta Islam.

**Writing – original draft:** Adel Md. Kibria, Md. Nazmul Islam, Md. Shaffiul Alam.

**Writing – review & editing:** Abdullah Abu Rafeh, RS Mahmud Hasan, Md. Siddiqur Rahman, A. K. M. Anisur Rahman.

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
