## [Decision Letter · Decision Letter 0]

5 Nov 2025

Dear Dr. Rahman,

Thank you for submitting your manuscript to PLOS ONE. After careful consideration, we feel that it has merit but does not fully meet PLOS ONE’s publication criteria as it currently stands. Therefore, we invite you to submit a revised version of the manuscript that addresses the points raised during the review process.

**ACADEMIC EDITOR:**

The paper has now been reviewed by an expert in the field. While the reviewers acknowledge that your study addresses an important gap — namely the absence of data on Mycobacterium avium subsp. paratuberculosis (MAP) in Bangladeshi dairy herds — they have identified several issues that must be resolved before the manuscript can be considered further. After careful consideration of the reviewer’s comments and my own evaluation, I am returning the manuscript for major revision. The reviewer appreciates the novelty and potential value of your work, but notes that the current version overstates its representativeness and certainty. Specifically, the survey’s non-random sampling, limited geographical balance, and use of milk ELISA only restrict the generalizability of the conclusions. Furthermore, several methodological details and interpretations require clarification or revision. Please submit your revised manuscript. Along with your revision, include a point-by-point response detailing how you have addressed each reviewer comment and where changes were made in the text.

We look forward to receiving your revised manuscript.

Kind regards,

Faham Khamesipour, Ph.D.

Academic Editor

PLOS ONE

“This research was funded by Livestock and Dairy Development Project, grant number 2022/7/LDDP.”

4. Thank you for stating the following in the Funding Section of your manuscript:

“This research was funded by Livestock and Dairy Development Project, grant number 2022/7/LDDP”

“This research was funded by Livestock and Dairy Development Project, grant number 2022/7/LDDP.”

6. We note that Figure 1 in your submission contain [map/satellite] images which may be copyrighted. All PLOS content is published under the Creative Commons Attribution License (CC BY 4.0), which means that the manuscript, images, and Supporting Information files will be freely available online, and any third party is permitted to access, download, copy, distribute, and use these materials in any way, even commercially, with proper attribution. For these reasons, we cannot publish previously copyrighted maps or satellite images created using proprietary data, such as Google software (Google Maps, Street View, and Earth). For more information, see our copyright guidelines: http://journals.plos.org/plosone/s/licenses-and-copyright.

7. We are unable to open your Supporting Information file [Supplementary_file_3_R_code_herd_level_risk_factors.Rmd, Supplementary-file_4_R_code_cow_level_risk_factors.Rmd and Supplementary_file_5_R_code_herd_cow_level_true_prevalence.Rmd]. Please kindly revise as necessary and re-upload.

Additional Editor Comments:

The paper has now been reviewed by an expert in the field. While the reviewers acknowledge that your study addresses an important gap — namely the absence of data on Mycobacterium avium subsp. paratuberculosis (MAP) in Bangladeshi dairy herds — they have identified several issues that must be resolved before the manuscript can be considered further. After careful consideration of the reviewer’s comments and my own evaluation, I am returning the manuscript for major revision. The reviewer appreciates the novelty and potential value of your work, but notes that the current version overstates its representativeness and certainty. Specifically, the survey’s non-random sampling, limited geographical balance, and use of milk ELISA only restrict the generalizability of the conclusions. Furthermore, several methodological details and interpretations require clarification or revision. Please submit your revised manuscript. Along with your revision, include a point-by-point response detailing how you have addressed each reviewer comment and where changes were made in the text.

Reviewers' comments:

Reviewer's Responses to Questions

**Comments to the Author**

1. Is the manuscript technically sound, and do the data support the conclusions?

Reviewer #1: Yes

Reviewer #2: Partly

2. Has the statistical analysis been performed appropriately and rigorously?

Reviewer #1: Yes

Reviewer #2: I Don't Know

3. Have the authors made all data underlying the findings in their manuscript fully available?

Reviewer #1: Yes

Reviewer #2: Yes

4. Is the manuscript presented in an intelligible fashion and written in standard English?

Reviewer #1: Yes

Reviewer #2: Yes

Reviewer #1: Title and Abstract

The phrase “True prevalence” could mislead readers to assume diagnostic certainty; consider “Estimated true prevalence.”

The abstract contains excessive numerical detail i.e. confidence intervals and regression coefficients, that might be summarized more succinctly for readability.

The term “nationwide” might be overstated given sampling in only 14 districts.

Introduction

This section provides a comprehensive background on Mycobacterium avium subsp. paratuberculosis (MAP), its global distribution, and regional context and clearly identifies the knowledge gap and study rationale for Bangladesh.

Materials and Methods

The sampling strategy for herd selection is described as “convenient,” which introduces potential bias and limits national representativeness.

The criteria for selecting 14 districts and seropositive herds for individual sampling are not fully explained.

The diagnostic sensitivity and specificity values for the ELISA (herd-level Se = 9.58%) seem unusually low; justification or source verification is necessary.

Results

Results are systematically presented from descriptive statistics to modeling outcomes. Figures and tables are well structured and appropriately referenced. The clear distinction between apparent and true prevalence is valuable.

The jump from apparent to true herd-level prevalence (7.2 % → 71.2 %) seems implausibly large; discussion of possible model over-correction or low test sensitivity should appear here.

Map need better captions specifying sample numbers per district.

Discussion

This section is well written and compares results effectively with studies from South Asia and other countries. Also appropriately discusses diagnostic limitations and implications for disease control. Integrates productivity and reproductive performance findings logically.

Reviewer #2: General comments:

The manuscript describes the outcome of a survey to evaluate the prevalence of paratb in milk samples in Bangladesh. Although the manuscript contains useful information, as nothing was known about the prevalence in the county before, it has some weaknesses which should be addressed before it can be considered for publication. The main issues are, that the survey was not statistically balanced and therefore does not reflect the situation in the whole country. Furthermore, the authors tend to overinterpret their results and the manuscript lacks a critically discussion, addressing the limitations of the study. Specific comments are provided below which should be addressed during a thorough review of the manuscript.

Specific comments:

Line 22: Why seroprevalence? Serum was not tested throughout the study.

Line 32-35: There were some more significant risk factors, please list them all

Line 63: “…limited sensitivity in eraly stages of the infection, meaning…”

Line 66: “…the paratuberculosis status and prevalence…”

Line 70: “…PCR (polymerase chain reaction).”

Line 71: “…ELISA (enzyme linked immunosorbent assay) …”

Line 71: specify “other tests”

Line 71: Why approximately? Be precise

Line 72: Replace “although” by “and” or “while”

Line 73: Please specify large, how many animals?

Line 77: What kind of skin test?

Line 96: Please rephrase, as farms >25 lactating cows still are small and should not be addressed as large-scale herds, although this might be the case for the situation in Bangladesh.

Line: Why was the expected prevalence 50%? Above it is stated that it is around 30-40% in other Asian countries. Please justify.

Line 108: How were the farms selected for the convenient sampling?

Line 109: If bulk milk samples were tested, why “seropositive” herds? Were there serum samples taken also? Please check and change throughout the manuscript.

Line 110: Provide more details of the farms and cows enrolled in the study, move lines 200-213 to this part of the manuscript, as they are a description of the study population and not a result of your study.

Line 110: Provide manufacturer for the software

Line 113: Each cow of the farm or from selected cows? Was this a combined ample of all for quarters?

Line 115: Department for Medicine of which institution?

Line 117: What are pretested questionnaires? How were they pretested?

Line 119: Purchase/trading of animas and contact to other herds (pasture etc.) are well known risk factors for the transmission of MAP. Why were they mot included in the survey?

Line 137: (ODNC)

Line 150: Where were the cut off values taken from? Manufacturer?

Line 163-164: Are these the Sp and Se values for milk? Please provide a reference, the values could not be found on the home page of the producer of the ELISA.

Line 202: Delete .0 for the mean of 119

Line 209: How were the sampled cows distributed to the farms? In the end, only 4 farms were included concerning single animal testing in the whole survey, although you identified 10 positive farms.

Line 211: please check your numbers, this is not possible: If there was a cow with 0 calves included, the lactation period can not range from 1-11 months.

Line 212: How was the body weight determined?

Line 221: Seroprevalence/Milk?

Line 222: In line 107-109 it is stated, that 10 out of 138 sampled herds were positive = 13.8%, here a prevalence of 7.2% is calculated, how? And why seroprevalence again? Was serum sampled or only milk?

Line 228: Seroprevalence?

Line 230: Seroprevalence/Milk?

Fig. 3: What does the line in the middle of the figures stand for? Add key

Fig. 5: Add key (SP), each figure hast to be self-explaining without connection to the text of the manuscript.

Line 267-269: This paragraph should be moved to the beginning of the results, as these are quite general outcomes of the study

Table 1: The variable “semen source” is not explained at any point of the manuscript. What is source 1, 2 etc.?

Table 1 and 2: Add key and level of significance

Line 310: This gap was not reveled by the study, this is obviously the case when diagnostic tests have a low sensitivity, delete statement.

Line 314-320: Please be more modest when interpreting the results of your study. The survey was not statistically balanced for the whole country, milk samples were used only (which are known not to be the best medium for Paratb detection), the herds showed a high variety in size. All this should be considered and discussed critically. Fig. 1 for example shows, that the herd prevalence was 0% in one province and 100% in a neighbor province-this does most probably not reflect the real situation but is caused by the study design.

Line 321-326: Same as above, consider and discuss weaknesses of the study and interpret with caution.

Line 339: There are also some studies showing, that cows with a high milk yield more often are MAP-positive, this also should be discussed.

Line 344-349: This should be discussed to a further extend, only because there is an association between two findings it doesn’t necessary imply that they are related. There is much more literature available concerning risk factors for Paratb in dairy cows. Also, some well known risk factors (purchase, contact to other herd) were no included in this study. Also, some found strong associations like age are not discussed at all.

Line 325-355: Not really a result of the study, delete. On the other hand, the main results (prevalence on the herd and individual level) should be added to the conclusions.

**Do you want your identity to be public for this peer review?** For information about this choice, including consent withdrawal, please see our Privacy Policy

Reviewer #1: **Yes:** Muhammad Avais

Reviewer #2: No

---

## [Author Response · Author response to Decision Letter 1]

24 Nov 2025

We are grateful for the valuable comments and suggestions from the academic editor and both reviewers, which have resulted in an enhanced manuscript. The modifications we implemented based on their feedback are indicated as tracked changes in the revised manuscript.

Academic Editor’s Summary

The paper has now been reviewed by an expert in the field. While the reviewers acknowledge that your study addresses an important gap — namely the absence of data on Mycobacterium avium subsp. paratuberculosis (MAP) in Bangladeshi dairy herds — they have identified several issues that must be resolved before the manuscript can be considered further. After careful consideration of the reviewer’s comments and my own evaluation, I am returning the manuscript for major revision. The reviewer appreciates the novelty and potential value of your work but notes that the current version overstates its representativeness and certainty. Specifically, the survey’s non-random sampling, limited geographical balance, and use of milk ELISA only restrict the generalizability of the conclusions. Furthermore, several methodological details and interpretations require clarification or revision.

Response: We have thoroughly addressed all possible reviewer and editorial concerns. The revised manuscript now clearly acknowledges the study’s limitations, such as the non-random sampling design, limited geographic representation, and the diagnostic scope of the milk ELISA, with appropriate caution in interpretation and generalizability. Methodological transparency has been improved by including detailed descriptions of the sampling rationale, herd selection criteria, laboratory procedures, and statistical analyses to ensure clarity and reproducibility. Interpretations and conclusions have been refined to moderate any overstatements of representativeness or certainty, ensuring that all claims are fully supported by the study design and data. Additionally, minor editorial and structural revisions were made throughout the manuscript to enhance overall readability, coherence, and scientific precision.

Reviewer #1

General Assessment

The study is valuable but needs clarification and moderation in claims.

Representativeness and diagnostic certainty are overstated.

The term “True prevalence” in the title may mislead readers.

Detailed Comments

Title and Abstract

The phrase “True prevalence” could mislead readers to assume diagnostic certainty; consider “Estimated true prevalence.”

Response: We have revised the term ‘true prevalence’ to ‘estimated true prevalence,’ following the reviewer’s recommendation..

The abstract contains excessive numerical detail i.e. confidence intervals and regression coefficients, that might be summarized more succinctly for readability.

Response: Excessive numerical details were summarized in the abstract.

The term “nationwide” might be overstated given sampling in only 14 districts.

Response: We have deleted the term ‘nationwide’ from abstract.

Introduction

This section provides a comprehensive background on Mycobacterium avium subsp. paratuberculosis (MAP), its global distribution, and regional context and clearly identifies the knowledge gap and study rationale for Bangladesh.

Materials and Methods

The sampling strategy for herd selection is described as “convenient,” which introduces potential bias and limits national representativeness.

Response: We agree with the reviewer that the purposive sampling strategy introduces potential bias and limits national representativeness. This limitation is now explicitly acknowledged in the Discussion, and Abstract, and we have clarified that the findings apply only to the study population and should not be interpreted as nationally representative.

The criteria for selecting 14 districts and seropositive herds for individual sampling are not fully explained.

Response: The criteria for selecting 14 districts and testing positive herds for individual sampling are described in the revised manuscript.

The diagnostic sensitivity and specificity values for the ELISA (herd-level Se = 9.58%) seem unusually low; justification or source verification is necessary.

Response: We have used the appropriate sensitivity (Se) and specificity (SP) of the ELISA utilized in this study and have reanalyzed the data.

Results

Results are systematically presented from descriptive statistics to modeling outcomes. Figures and tables are well structured and appropriately referenced. The clear distinction between apparent and true prevalence is valuable.

The jump from apparent to true herd-level prevalence (7.2 % → 71.2 %) seems implausibly large; discussion of possible model over-correction or low-test sensitivity should appear here.

Response: We have reanalyzed the data using new and suitable prior information and have provided a revised description of the true estimated prevalence in the manuscript..

Map need better captions specifying sample numbers per district.

Response: We thank the reviewer for pointing this out. The figure caption has been revised to clarify that n refers to the number of herds sampled in each district, rather than the number of individual animals.

Discussion

This section is well written and compares results effectively with studies from South Asia and other countries.

Also appropriately discusses diagnostic limitations and implications for disease control.

Response: We thank the reviewer for this suggestion. In response, we have revised the discussion to explicitly address the diagnostic limitations of ELISA, including its imperfect sensitivity and specificity, which can lead to underestimation of true herd-level MAP prevalence. We have also highlighted the implications for disease control, emphasizing that accurate prevalence estimation through standardized protocols and adjusted or complementary testing is essential for effective surveillance and intervention.

Integrates productivity and reproductive performance findings logically.

Response: We thank the reviewer for this suggestion. The discussion has been revised to better integrate findings on productivity and reproductive performance. We now emphasize that MAP seropositivity is associated with modest reductions in milk yield as well as increased odds of reproductive disorders, and we discuss these outcomes in a biologically plausible context. Additionally, we clarify that these findings represent associations rather than causal relationship.

Reviewer #2 Comments

General Assessment

The manuscript describes the outcome of a survey to evaluate the prevalence of paratb in milk samples in Bangladesh. Although the manuscript contains useful information, as nothing was known about the prevalence in the county before, it has some weaknesses which should be addressed before it can be considered for publication. The main issues are, that the survey was not statistically balanced and therefore does not reflect the situation in the whole country. Furthermore, the authors tend to overinterpret their results and the manuscript lacks a critically discussion, addressing the limitations of the study. Specific comments are provided below which should be addressed during a thorough review of the manuscript.

Specific Comments (by line number)

Abstract & Introduction

Line 22: Why seroprevalence? Serum was not tested throughout the study.

Response: We appreciate the reviewer’s careful observation. The term seroprevalence has been replaced with antibody prevalence throughout the manuscript to accurately reflect that milk, rather than serum, was tested for antibodies against Mycobacterium avium subsp. paratuberculosis.

Line 32-35: There were some more significant risk factors, please list them all

Response: We thank the reviewer for the comment. Only variables significant in the final multivariable models are reported as key results (Table 2 for herd-level, Table 4 for cow-level). Variables associated in univariable screening are part of the model-building process and are listed in Table 1 and Table 3 for transparency but are not described as final findings.

Line 63: „…limited sensitivity in early stages of the infection, meaning…”

Response: Thank you for your edit. We have made the suggested modifications.

Line 66: “…the paratuberculosis status and prevalence…”

Response: Thank you for your edit; we have made the suggested modifications..

Line 70: “…PCR (polymerase chain reaction).”

Response: Elabortaed as suggested.

Line 71: “…ELISA (enzyme linked immunosorbent assay) …”

Response: Elabortaed as suggested.

Line 71: specify “other tests”

Response: Other test was specified.

Line 71: Why approximately? Be precise

Response: We have deleted the word approximately.

Line 72: Replace “although” by “and” or “while”

Response: We removed that portion of the sentence.

Line 73: Please specify large, how many animals?

Response: We have rephrased the sentence.

Line 77: What kind of skin test?

Response: We have replaced ‘skin test’ with the delayed-type hypersensitivity test.

Line 96: Please rephrase, as farms >25 lactating cows still are small and should not be addressed as large-scale herds, although this might be the case for the situation in Bangladesh.

Response: We have revised the statement based on the reviewer’s suggestion..

Line: Why was the expected prevalence 50%? Above it is stated that it is around 30-40% in other Asian countries. Please justify.

Response: Although published studies from parts of Asia report MAP seroprevalence around 30–40%, we used an expected prevalence of 50% in the sample-size formula because this value maximizes the binomial variance p(1-p) and therefore yields the largest (most conservative) sample size for a given precision. Using the same precision (d = 0.035), the required sample size would have been 784 animals with p=0.50, whereas using a more realistic estimate of 35% would have produced a smaller requirement of approximately 715 animals. Because we ultimately collected 880 samples, our achieved sample size exceeds both estimates, ensuring greater statistical power, narrower confidence intervals, and sufficient coverage for subgroup analyses. Thus, the final sample size remains more than adequate regardless of whether 35% or 50% is used as the expected prevalence.

Line 108: How were the farms selected for the convenient sampling?

Response: The herd selection strategy has been described in the revised manuscript.

Line 109: If bulk milk samples were tested, why “seropositive” herds? Were there serum samples taken also? Please check and change throughout the manuscript.

Response: We have changed seropositive to LEISA positive herds.

Line 110: Provide more details of the farms and cows enrolled in the study, move lines 200-213 to this part of the manuscript, as they are a description of the study population and not a result of your study.

Response: The farms and cow enrollment have been described in detail.

Line 110: Provide manufacturer for the software

Response: Manufacturer detail provided

Line 113: Each cow of the farm or from selected cows? Was this a combined ample of all for quarters?

Response: We thank the reviewer for this comment. We confirm that milk samples were collected from each individual cow, not from a subset of animals. For each cow, a composite sample was taken by pooling milk from all four quarters. The sentence in the Methods section has been revised accordingly.

Line 115: Department for Medicine of which institution?

Response: The Institute of the department added in the revised manuscript.

Line 117: What are pretested questionnaires? How were they pretested?

Response: We appreciate the reviewer’s query. By “pretested questionnaires,” we refer to structured questionnaires that were piloted before the main survey to ensure clarity, relevance, and consistency. The questionnaires were tested on a small number of farms and animals outside the study population. Based on feedback from farmers and field staff, several questions were refined to improve wording and reduce ambiguity. This has now been clarified in the revised Methods section.

Line 119: Purchase/trading of animas and contact to other herds (pasture etc.) are well known risk factors for the transmission of MAP. Why were they mot included in the survey?

Response: Thank you for this valuable comment. All enrolled herds practiced zero-grazing and were managed entirely indoors, eliminating inter-herd contact through pasture. Furthermore, reliable records on animal purchase or trading were not available. For these reasons, these variables could not be included in the survey. This clarification has been added to the revised manuscript.

Line 137: (ODNC)

Response: Thank you, corrected.

Line 150: Where were the cut off values taken from? Manufacturer?

Response: Yes, the cutoff value was taken from the manufacturer.

Line 163-164: Are these the Sp and Se values for milk? Please provide a reference, the values could not be found on the home page of the producer of the ELISA.

Response: Thank you for your comment. During the initial submission, we could not find published herd- or cow-level sensitivity (Se) and specificity (Sp) values for the iELISA used to detect antibodies in bulk or individual cow milk. Although we contacted the manufacturer, the values provided were unpublished. Following your suggestion, we identified two published studies reporting these diagnostic performance estimates: for herd-level prevalence, we used an iELISA sensitivity of 63% (95% CI: 41–81%) and specificity of 92% (95% CI: 86–96%) [Reference 20]; for cow-level prevalence, we used a commercial ELISA sensitivity of 85.7% (95% CI: 76.4–95.1%) and specificity of 95.1% (95% CI: 92.6–97.7%) [Reference 21]. Using this prior information, we have updated and redone the estimation of true antibody prevalence.

Line 202: Delete .0 for the mean of 119

Response: Deleted as suggested

Line 209: How were the sampled cows distributed to the farms? In the end, only 4 farms were included concerning single animal testing in the whole survey, although you identified 10 positive farms.

Response: Thank you for your comment. To clarify, ten herds were initially identified as ELISA-positive. From these, 880 cows were randomly selected using the ‘RANDBETWEEN’ function in Excel to ensure unbiased selection. However, the inclusion of herds for individual cow-level testing was limited by herd owner willingness and logistical feasibility, resulting in only four herds contributing cows for individual sampling. Both herd-level bulk milk testing and cow-level individual testing were used in this study, and we have revised the manuscript to clearly explain the distribution of sampled cows across herds.

Line 211: please check your numbers, this is not possible: If there was a cow with 0 calves included, the lactation period can not range from 1-11 months.

Response: We thank the reviewer for pointing this out. Upon careful review, we identified a typographical error in the number of calves recorded for one cow. This error led to an incorrect representation of the lactation period range. The data have now been corrected, and the summary statistics have been updated accordingly.

Line 212: How was the body weight determined?

Response: We appreciate the reviewer’s query. The body weights reported in our study were obtained from the herd records. Herds use either digital weighing scales or other standard on-farm methods to measure the body weight of cows, and these measurements were recorded as part of their routine management practices.

Line 221: Seroprevalence/Milk?

Response: All instances of “seroprevalence” in the manuscript have been updated to “antibody prevalence” in the revised version to accurately reflect the measured outcome.

Line 222: In line 107-109 it is stated, that 10 out of 138 sampled herds were positive = 13.8%, here a prevalence of 7.2% is calculated, how? And why seroprevalence again? Was serum sampled or only milk?

Response: The calculation of herd-level prevalence is correct: 10 positive herds out of 138 sampled corresponds to 7.2% herd-level prevalence. Additionally, in the revised manuscript, all instances of “seroprevalence” have been replaced with “antibody prevalence” to accurately reflect that only milk samples we

---

## [Decision Letter · Decision Letter 1]

9 Dec 2025

Estimated true prevalence and associated risk factors of bovine paratuberculosis antibodies in dairy herds in Bangladesh

PONE-D-25-53226R1

Dear Dr. Rahman,

We’re pleased to inform you that your manuscript has been judged scientifically suitable for publication and will be formally accepted for publication once it meets all outstanding technical requirements.

Kind regards,

Faham Khamesipour, Ph.D.

Academic Editor

PLOS One

Additional Editor Comments (optional):

Reviewers' comments:

Reviewer's Responses to Questions

**Comments to the Author**

Reviewer #1: All comments have been addressed

Reviewer #2: All comments have been addressed

2. Is the manuscript technically sound, and do the data support the conclusions?

Reviewer #1: Yes

Reviewer #2: Yes

3. Has the statistical analysis been performed appropriately and rigorously?

Reviewer #1: Yes

Reviewer #2: I Don't Know

4. Have the authors made all data underlying the findings in their manuscript fully available?

Reviewer #1: Yes

Reviewer #2: Yes

5. Is the manuscript presented in an intelligible fashion and written in standard English?

Reviewer #1: Yes

Reviewer #2: Yes

Reviewer #1: (No Response)

Reviewer #2: The authors performed a thorough and detailed revision of the manuscript, addressing all questions pointed out at the first revision, This markedly improved the quality of the manuscript!

**Do you want your identity to be public for this peer review?** For information about this choice, including consent withdrawal, please see our Privacy Policy

Reviewer #1: **Yes:** Muhammad Avais

Reviewer #2: No

---

## [Editor Report · Acceptance letter]

PONE-D-25-53226R1

PLOS One

Dear Dr. Rahman,

I'm pleased to inform you that your manuscript has been deemed suitable for publication in PLOS One. Congratulations! Your manuscript is now being handed over to our production team.

Kind regards,

on behalf of

Dr. Faham Khamesipour

Academic Editor

PLOS One